# Multi-Omics Analysis of Diabetic Heart Disease in the *db/db* Model Reveals Potential Targets for Treatment by a Longevity-Associated Gene

**DOI:** 10.3390/cells9051283

**Published:** 2020-05-21

**Authors:** Ashton Faulkner, Zexu Dang, Elisa Avolio, Anita C Thomas, Thomas Batstone, Gavin R Lloyd, Ralf JM Weber, Lukáš Najdekr, Andris Jankevics, Warwick B Dunn, Gaia Spinetti, Carmine Vecchione, Annibale A Puca, Paolo Madeddu

**Affiliations:** 1Bristol Medical School: Translational Health Sciences, University of Bristol, Bristol BS2 8HW UK; ashton.faulkner@bristol.ac.uk (A.F.); zexudang@googlemail.com (Z.D.); elisa.avolio@bristol.ac.uk (E.A.); a.thomas@bristol.ac.uk (A.C.T.); 2School of Biological Sciences, University of Bristol, Bristol BS1 5QD, UK; tom.batstone@bristol.ac.uk; 3Phenome Centre Birmingham, University of Birmingham, Birmingham B15 2TT, UK; g.r.lloyd@bham.ac.uk (G.R.L.); r.j.weber@bham.ac.uk (R.J.M.W.); l.najdekr@bham.ac.uk (L.N.); a.jankevics@bham.ac.uk (A.J.); w.dunn@bham.ac.uk (W.B.D.); 4School of Biosciences, University of Birmingham, Birmingham B15 2TT, UK; 5Institute of Metabolism and Systems Research, University of Birmingham, Birmingham, B15 2TQ, UK; 6IRCCS Multimedica, Cardiovascular Department, 20138 Milan, Italy; gaia.spinetti@multimedica.it (G.S.); apuca@unisa.it (A.A.P.); 7Department of Medicine, University of Salerno, 84081 Baronissi, Italy; cvecchione@unica.it; 8IRCCS Neuromed, Department of Vascular Physiopathology, 86077 Pozzilli, Italy

**Keywords:** cardiomyopathy, type-2 diabetes, longevity, gene therapy, BPIFB4

## Abstract

Characterisation of animal models of diabetic cardiomyopathy may help unravel new molecular targets for therapy. Long-living individuals are protected from the adverse influence of diabetes on the heart, and the transfer of a longevity-associated variant (LAV) of the human *BPIFB4* gene protects cardiac function in the *db/db* mouse model. This study aimed to determine the effect of *LAV-BPIFB4* therapy on the metabolic phenotype (ultra-high-performance liquid chromatography-mass spectrometry, UHPLC-MS) and cardiac transcriptome (next-generation RNAseq) in *db/db* mice. UHPLC-MS showed that 493 cardiac metabolites were differentially modulated in diabetic compared with non-diabetic mice, mainly related to lipid metabolism. Moreover, only 3 out of 63 metabolites influenced by *LAV-BPIFB4* therapy in diabetic hearts showed a reversion from the diabetic towards the non-diabetic phenotype. RNAseq showed 60 genes were differentially expressed in hearts of diabetic and non-diabetic mice. The contrast between *LAV-BPIFB4*- and vehicle-treated diabetic hearts revealed eight genes differentially expressed, mainly associated with mitochondrial and metabolic function. Bioinformatic analysis indicated that *LAV-BPIFB4* re-programmed the heart transcriptome and metabolome rather than reverting it to a non-diabetic phenotype. Beside illustrating global metabolic and expressional changes in diabetic heart, our findings pinpoint subtle changes in mitochondrial-related proteins and lipid metabolism that could contribute to *LAV-BPIFB4*-induced cardio-protection in a murine model of type-2 diabetes.

## 1. Introduction

Type-2 diabetes mellitus is a chronic metabolic disorder that continues to rise in prevalence and significantly increases the risk of developing cardiovascular complications. One such complication is diabetic cardiomyopathy, a specific form of cardiac disease that evolves independently of coronary artery disease, hypertension or valvular disease [1,2]. Although the specific cellular and molecular mechanisms of diabetic cardiomyopathy remain relatively unknown, the diabetic heart exhibits typical features of accelerated cellular ageing, namely impaired metabolic flexibility and mitochondrial dysfunction, endoplasmic reticulum stress, altered Ca^2+^ handling, and disruption in redox balance [1,2]. As no specific therapy is available to halt the progression of diabetic cardiomyopathy towards heart failure, there is an urgent need for a new therapeutic approach [3,4].

Long-living individuals (LLIs) appear to be relatively resistant to such diseases and the genetics of healthy longevity may, therefore, offer the opportunity to develop novel therapies [5,6,7]. A genome-wide association study on LLIs of three different geographical areas identified a Longevity Associated Variant (LAV) of the bactericidal/permeability-increasing fold-containing family B member 4 (BPIFB4), a four-missense single-nucleotide polymorphism haplotype allele. BPIFB4 belongs to a family of proteins with lipid binding pockets involved in many functions, from anti-microbial (BPI) to cholesterol handling (CETP). Specifically, LAV confers novel functions to BPIFB4 protein, which becomes able to activate eNOS via a SDF1/CXCR4/calcium mediated mechanism. Additionally, we have described a unique therapeutic effect of the LAV isoform as compared to the wild type (WT). Indeed, delivery of the LAV, but not the WT form of *BPIFB4* gene effectively contrasts the development of atherosclerosis and benefits the recovery from ischaemia by potentiating endothelial function [7,8]. These effects are blunted by CXCR4 inhibitors and are associated with a monocyte skewing toward an M2 phenotype. Furthermore, we have recently published data demonstrating the protective effect of *LAV-BPIFB4* from the progressive decline in cardiac function in the leptin receptor-mutant *db/db* mouse, a well-used animal model of obesity-related type-2 diabetes [9]. Alongside the improvement in cardiac contractility, we observed a significant reduction in intracellular lipid accumulation without a restoration in glucose uptake, suggesting a positive effect of LAV-BPIFB4 on cardiomyocyte lipid handling and/or mitochondrial function. While the over-reliance and accumulation of lipid, together with the progressive decline in cardiac function, is well-described in the *db/db* model [10,11,12,13], a wider assessment of cardiac metabolic and transcriptomic changes is limited within the available literature [14,15]. Furthermore, to the best of our knowledge, there are no multi-omics-based reports examining new experimental treatments that halt the progression of the disease.

Although the existence of an animal model that truly captures every aspect of the human condition is not possible, better characterisation and understanding of available models has been recognised as a significant factor in helping to facilitate the development of novel therapies [11,16]. Therefore, in this study, we extended our previous analysis and performed a metabolomic and transcriptomic investigation to further characterise changes within the heart of diabetic *db/db* mice and identify potential changes exerted by our novel longevity-associated gene (*LAV-BPIFB4*) therapy.

## 2. Materials and Methods

### 2.1. Ethics

Experiments were performed according to a randomised protocol under ethical license from the British Home Office (PPL number 30/3373), the IRCCS Neuromed Animal Care Review Board, and the Istituto Superiore di Sanità, Rome (1163/2015- PR). All procedures were compliant with the EU Directive 2010/63/EU and principles stated in the Guide for the Care and Use of Laboratory Animals (The Institute of Laboratory Animal Resources, 1996).

### 2.2. Viral Vector

Adeno-associated virus (AAV-9) containing the human *LAV-BPIFB4* sequence was produced as described previously [7]. For each viral preparation, physical titres (GC/mL) were determined by averaging the titre achieved by dot-blot analysis and by PCR quantification using TaqMan20 (Applied Biosystems, Carlsbad, CA, USA).

### 2.3. In Vivo Protocol

Nine-week old male C57BLKS/J-Leprdb/Leprdb/Dock7+ [*db/db*] mice (Envigo, Bicester, Oxfordshire, UK) under isoflurane anaesthesia were injected via the tail vein with either AAV9-*LAV-BPIFB4* (100 µL at 1 × 10^12^ GC/mL) or vehicle control (100 µL AAV-GFP at 1 × 10^12^ GC/mL or 100 µL PBS). Diabetes was confirmed by regular measurements of glycosuria, as previously described [9] . Age- and sex-matched lean non-diabetic mice (C57BLKS/J-Leprdb/LeprWT Dock7+ (*wt/db*); Envigo) were used as controls. Mice were terminated 4 weeks post-intervention and hearts were collected and processed for molecular and metabolomic analyses. The expression of BPIFB4 within the heart was confirmed by immunohistochemistry, using an antibody that recognised both the human and murine protein (GeneTex, GTX51455, 1:100) (Appendix A). Additional information about the organ tropism of the AAV9, and pattern of cardiac BPIFB4 expression, can be found in our recently published study [9].

### 2.4. Next-Generation RNA Sequencing

Studies were performed in left ventricular samples from 3 to 4 hearts per group. Libraries for RNA-seq were prepared using the Illumina TruSeq Stranded Total RNA protocol and sequenced on the Illumina NextSeq 4000 to generate approximately 40 million reads per sample. Data was quality assessed using FastQC v0.11.5 (https://www.bioinformatics.babraham.ac.uk/projects/fastqc/), and reads were trimmed to remove adapters and to attain a minimum Phred quality of 20 using Trimmomatic v0.36 [17]. Analysis of differential gene expression was carried out using the Partek Flow GSA method, and resulting gene lists filtered to a false discovery rate corrected *p*-value (FDR) of 0.05. Blast2GO was used to perform the GO enrichment test.

### 2.5. RNA Isolation and RT-qPCR

Total RNA was isolated from flash-frozen murine left ventricular myocardium using TRIzol (ThermoFisher Scientific, Paisley, UK) according to the manufacturer's instructions, and reverse transcribed using the High-Capacity RNA-to-cDNA kit (Thermo Fisher Scientific, Paisley, UK). Oligonucleotide primer sequences were designed using Primer3 and NCBI primer-BLAST software, or obtained from published literature, and synthesised by Eurofins MWG Operon (Wolverhampton, UK) (see Appendix A for sequences). TaqMan primer-probes were obtained from ThermoFisher Scientific (Paisley, UK). Analysis was performed using the 2^−ΔΔCt^ method with results normalised to β-actin (*ACTB*) internal control gene.

### 2.6. Protein Extraction and Western Blotting

Western blot analysis was conducted on six left ventricles per group. Protein concentration was determined using the BCA Protein Assay Kit (Thermo Fisher Scientific, Paisley, UK). Primary antibodies (listed in Appendix A) were incubated for 16 h at 4 °C. β-tubulin was used as loading control. Anti-rabbit IgG or anti-mouse IgG HRP-conjugated were employed as secondary antibodies (both 1:5000; GE Healthcare, Amersham, UK). Membrane development was performed by an enhanced chemiluminescence-based detection method (ECL™ Prime Western Blotting Detection Reagent; GE Healthcare Amersham, UK) in a ChemiDoc-MP system (Bio-Rad, Watford, UK). No more than one stripping procedure was performed on an individual membrane (Restore^TM^ Plus Western Blot Stripping Buffer; Thermo Fisher Scientific, Paisley, UK). Two samples were loaded on each gel from each experimental group and quantitative comparisons performed between samples on the same blot.

### 2.7. Untargeted Metabolomics–Ultra-High-Performance Liquid Chromatography-Mass Spectrometry (UHPLC-MS) 

Water-soluble metabolites and lipids were extracted from pre-weighed tissues (*n* = 5 to 7 whole hearts per group) using a two-step Bligh and Dyer style biphasic extraction protocol with some modifications [18,19]. Tissue samples were cut into approximately 30 mg pieces, accurately weighed, transferred into Precellys tubes (Precellys, CK14; Stretton Scientific, Stretton, UK) and homogenised using 15 μL ice-cold methanol per mg of tissue (LCMS grade, HiPerSolv; VWR, Lutterworth, UK) and 6 μL ice-cold water per mg of tissue (LCMS grade, LiChrosolv; Merck, Feltham, UK) in a bead-based homogenizer (Precellys 24; Stretton Scientific, Stretton, UK) with 2 × 10 sec bursts at 6400 rpm and at room temperature. The homogenate was transferred to a 1.8 mL glass vial, and 15 μL ice-cold chloroform per mg of tissue (HPLC grade, HiPerSolv CHROMANORM; VWR, Lutterworth, UK) and 7.5 μL ice-cold water per mg of tissue were added before vortexing (60 sec). The final ratio of solvents in the glass vials was 2:2:1.8 methanol:chloroform:water (v/v). Samples were set on ice (10 min), centrifuged (4000 g; 4 °C; 10 min) and left at room temperature for 10 min. The aliquots of water-soluble (500 μL) and non-polar (250 μL) phases were removed using a Hamilton syringe. Polar extracts were dried using a SpeedVac (Savant SPD111V; Thermo Fisher Scientific, Paisley, UK) and vapour trap (RVT5105230; Thermo Fisher Scientific, Paisley, UK), and non-polar extracts were dried under a nitrogen stream (Techne Sample Concentrator FSC400D; Thermo Fisher Scientific, Paisley, UK). Dried extracts were stored at −80 °C until analysis. Extract blanks were prepared by the same methods in the absence of tissue.

Water-soluble extracts were re-suspended in 170 μL of 3:1 acetonitrile/water (v/v) (LCMS grade, HiPerSolv; VWR, Lutterworth, UK / LCMS grade, LiChrosolv; Merck, Feltham, UK, respectively) and non-polar extracts were re-suspended in 170 μL of 3:1 propan-2-ol/water (v/v) (LCMS grade, LiChrosolv; Merck, Feltham, UK). Following resuspension solvent addition, samples were vortexed (30 sec). Separately for polar and non-polar samples, a pooled QC sample was prepared by combining 60 μL aliquots from re-suspended extracts and vortexing for 5 min. All samples were centrifuged (20,000 g; 4 °C; 20 min) and loaded into HPLC vials (VI-04-12-02RVG 300μL Plastic; Chromatography Direct, Runcorn, UK).

The samples were analysed applying two ultrahigh performance liquid chromatography-mass spectrometry (UHPLC-MS) methods using a Dionex UltiMate 3000 Rapid Separation LC system (Thermo Fisher Scientific, Paisley, UK) coupled with an electrospray Q Exactive Focus mass spectrometer (Thermo Fisher Scientific, Paisley, UK). Polar extracts were analysed on Accucore-150-Amide-HILIC column (100 × 2.1 mm, 2.6 μm; Thermo Fisher Scientific, Paisley, UK)–HILIC method. Mobile phase A consisted of 10 mM ammonium formate and 0.1% formic acid in 95% acetonitrile/water and mobile phase B consisted of 10 mM ammonium formate and 0.1% formic acid in 50% acetonitrile/water. Flow rate was set for 0.50 mL.min-1 with the following gradient: t = 0.0, 1% B; t = 1.0, 1% B; t = 3.0, 15% B; t = 6.0, 50% B; t = 9.0, 95% B; t = 10.0, 95% B; t = 10.5, 1% B; t = 14.0, 1% B, all changes were linear with curve = 5. The column temperature was set to 35 °C, and the injection volume was 2 μL. Data were acquired in positive and negative ionization modes separately within the mass range of 70–1050 m/z at resolution 70,000 (FWHM at m/z 200). Ion source parameters were set as follows: Sheath gas = 53 arbitrary units, Aux gas = 14 arbitrary units, Sweep gas = 3 arbitrary units, Spray Voltage = 3.5kV, Capillary temp. = 269 °C, Aux gas heater temp. = 438 °C. Data-dependent MS2 in ‘Discovery mode’ was used for the MS/MS spectra acquisition using following settings: resolution = 17,500 (FWHM at m/z 200); Isolation width = 3.0 m/z; stepped normalized collision energies (stepped NCE) = 25, 60, 100%. Spectra were acquired in three different mass ranges: 50–200 m/z; 200–400 m/z; 400–1000 m/z. Non-polar extracts were analysed on Hypersil GOLD column (100 × 2.1mm, 1.9 µm; Thermo Fisher Scientific, MA, USA)–LIPIDS method. Mobile phase A consisted of 10 mM ammonium formate and 0.1% formic acid in 60% acetonitrile/water and mobile phase B consisted of 10 mM ammonium formate and 0.1% formic acid in 90% propan-2-ol/water. Flow rate was set for 0.40 mL.min-1 with the following gradient: t = 0.0, 20% B; t = 0.5, 20% B, t = 8.5, 100% B; t = 9.5, 100% B; t = 11.5, 20% B; t = 14.0, 20% B, all changes were linear with curve = 5. The column temperature was set to 55 °C and the injection volume was 2 μL. Data were acquired in positive and negative ionization mode separately within the mass range of 150–2000 m/z at resolution 70,000 (FWHM at m/z 200). Ion source parameters were set as follows: Sheath gas = 50 arbitrary units, Aux gas = 13 arbitrary units, Sweep gas = 3 arbitrary units, Spray Voltage = 3.5kV, Capillary temp. = 263 °C, Aux gas heater temp. = 425 °C. Data dependent MS2 in ‘Discovery mode’ was used for the MS/MS spectra acquisition using following settings: resolution = 17,500 (FWHM at m/z 200); Isolation width = 3.0 m/z; stepped normalized collision energies (stepped NCE) = 20, 50, 80%. Spectra were acquired in three different mass ranges: 200–400 m/z; 400–700 m/z; 700–1500 m/z. A Thermo ExactiveTune 2.8 SP1 build 2806 was used as an instrument control software in both cases and data were acquired in profile mode. QC samples were analysed as the first ten injections and then every sixth injection with two QC samples at the end of the analytical batch. Two blank samples were analysed, the first as the 6th injection and then at the end of the batch.

Data acquired were converted from the instrument-specific format to the mzML file format applying the open access ProteoWizard software [20]. Deconvolution was performed with XCMS software [21] according to the following settings of Min peak width (4 for HILIC and 6 for LIPIDS); max peak width (30); ppm (12 for HILIC and 14 for LIPIDS); mzdiff (0.001); gapInit (0.5 for HILIC and 0.4 for LIPIDS); gapExtend[21]; bw (0.25); mzwid (0.01). A data matrix of metabolite features (m/z-retention time pairs) vs. samples was constructed with peak areas provided where the metabolite feature was detected for each sample. Putative annotation of metabolites or metabolite groups was performed by applying the PUTMEDID-LCMS workflows operating in the Taverna workflow environment [22]. We applied 12 ppm mass error for HILIC and 14 ppm mass error for LIPIDS data and a retention time range of 2 s in feature grouping and molecular formula and metabolite matching. As different metabolites can be detected with the same accurate m/z (for example, isomers with the same molecular formula), multiple annotations could be observed for a single detected metabolite feature. Also, a single metabolite could be detected as multiple molecules, particularly as a different type of ion (e.g., protonated and sodiated ions). All molecules were annotated according to guidelines for reporting of chemical analysis results, specifically to Metabolomics Standards Initiative level 2 [23].

A quality assurance and quality control (QA/QC) assessment was performed to measure drift across retention time, m/z and signal intensity and identify potential outliers. The first five QCs were used to equilibrate the analytical system and therefore subsequently removed from the data before the data was analysed. Principal Components Analysis (PCA) was performed to assess the technical variability (measured by the replicate analysis of a pooled QC sample) and biological variability as part of the quality control process. Prior to PCA, missing values in the data were replaced by applying k nearest neighbour (kNN) missing value imputation (k = 5) followed by probabilistic quotient normalization (PQN) [24], and glog transformation prior to data analysis. The data from the pooled QC samples were applied to perform QC filtering. For each metabolite feature detected QC samples 1-5 were removed and the relative standard deviation and percentage detection rate were calculated using the remaining QC samples. Blank samples at the start and end of a run were used to remove features from non-biological origins. Any feature with an average QC intensity less than 20 times the average intensity of the blanks was removed. Any sample with >50% missing values was excluded from further analysis. Metabolite features with an RSD > 30% and present in less than 90% of the QC samples were deleted from the dataset. Features with a <50% detection rate over all samples were also removed. Univariate analysis was performed applying t-tests on data normalised to total peak area per sample and glog transformed data with a critical *p*-value >0.05 applied. Fold changes were calculated using data normalised to total peak area per sample.

### 2.8. Statistical Analysis

Data represent means ± standard error of mean (SEM) of independent samples. For RT-qPCR and western blot analysis, differences between groups was analysed by Mann-Whitney U test. Significance was assumed when *p* = <0.05.

## 3. Results

### 3.1. Cardiac Lipid Metabolism is Altered in db/db Mouse Hearts with Further Changes Imparted by LAV-BPIFB4 Therapy

In 13-week old mice, untargeted UHPLC-MS metabolic phenotyping analysis of heart samples revealed 493 metabolites were differentially modulated in GFP-vehicle treated diabetic compared with GFP-vehicle treated non-diabetic hearts (*p* < 0.05; 0.8 < relative concentration fold change > 1.2). The major contributing metabolite classes included glycerophospholipids, triacylglycerides, ceramides and sphingolipids, acyl carnitines, fatty acids, diacylglycerides, cardiolipins, and peptides. Furthermore, when comparing *LAV-BPIFB4*-treated with *GFP*-vehicle treated diabetic *db/db* mice (*p* < 0.05; 0.8 < relative concentration fold change > 1.2), 63 metabolites were differentially modulated, of which 60 increased in the *LAV-BPIFB4*-treated group (Appendix A). Interestingly, only 3 metabolites (two cardiolipins and one glycerophospholipid) showed a reversion from the diabetic towards the non-diabetic phenotype following treatment (Appendix A).

Together, these data indicate significant remodelling of lipid metabolism within the GFP-vehicle treated diabetic *db/db* mouse heart, with a small additional remodelling (rather than reversal to non-diabetic) occurring in mice receiving *LAV-BPIFB4* therapy.

### 3.2. Changes in Cardiac Transcriptome of db/db Mice and Effect of LAV-BPIFB4 Therapy

As the metabolomic analysis indicated alterations in lipid metabolism, we next sought to identify if these changes were associated with alterations in the cardiac transcriptome.

RNAseq showed that 60 genes were differentially expressed (minimum 2-fold change) within the hearts of PBS-vehicle treated diabetic *db/db* mice compared with non-diabetic controls (40 up-regulated and 20 down-regulated) (Figure 1A,B and Appendix A). However, 266 genes were differentially expressed in the contrast between *LAV-BPIFB4*-treated diabetic and untreated non-diabetic mice (121 up-regulated and 145 down-regulated) (Figure 1C,D and Appendix A), with 56 of these genes being shared by the two contrasts. Interestingly, the Principal Component Analysis (PCA) showed that *LAV-BPIFB4*-treated diabetic mice diverged further from the non-diabetic phenotype than did the PBS-vehicle treated diabetic mice, especially along the PC1 axis (Figure 1E).

When comparing diabetic *db/db* mice receiving *LAV-BPIFB4* therapy with those receiving PBS-vehicle alone, eight genes (five up-regulated and three down-regulated) were differentially expressed when considering a minimum 2-fold change (Figure 1F,G), with this increasing to 20 genes (14 up-regulated and 6 down-regulated) at 1.5-fold change (Figure 1H and Appendix A). GOSlim classification identified that four differentially expressed genes were associated with mitochondrial function and 3 with metabolic function. Finally, a list of differentially expressed genes with no fold change cut-off (but retaining the FDR 0.05) was generated, which consisted of 64 genes (15 up-regulated and 49 down-regulated) (Appendix A). GOSlim classification identified 25 GO terms, of which 14 were associated with GO:0044255 (cellular lipid metabolic process), 15 to GO:0006629 (lipid metabolic process) and 6 to GO:0019395 (fatty acid oxidation). Analysis of the down-regulated genes alone yielded no significant enrichment results, while that of the up-regulated genes identified 23 GO terms, most being metabolic (17) and, more specifically, related to lipid metabolism (10).

To validate the RNAseq data, RT-qPCR analysis of 12 differentially expressed genes identified from the *LAV-BPIFB4* / PBS vehicle contrast (1.5-fold cut-off) was performed on the same samples, plus 3 additional hearts (totalling 6 samples per group) (Appendix A). This validation confirmed 10 out of the 12 genes were differentially modulated by *LAV-BPIFB4* in diabetic *db/db* hearts, of which 8 in a concordant direction between RNAseq and RT-qPCR (Figure 2). In line with the PCA, *LAV-BPIFB4* treatment further enhanced the diabetic expressional phenotype with respect to the non-diabetic control group when assessed by RT-qPCR.

### 3.3. Focused Expression Analysis of Mitochondrial-Related Pathways

Cardiomyocyte lipid accumulation can occur due to an elevation in lipid supply and induction of mitochondrial dysfunction [1]. Given our previous findings that *LAV-BPIFB4* therapy reduces lipid accumulation within the *db/db* heart, together with the altered lipid metabolism revealed by the metabolomic analysis reported in the present study, and further indicated by the sequencing annotations, we performed a targeted expression analysis of key proteins of mitochondrial-related pathways (Figure 3 and Appendix A).

PBS-vehicle-treated *db/db* hearts had increased mRNA levels of pyruvate dehydrogenase kinase 4 (*Pdk4*), a regulator of glucose oxidation (Figure 3A). Several genes relating to enzymes involved in fatty acid metabolism (carnitine palmitoyltransferase 1B (*Cpt1b*); medium-chain acyl-CoA dehydrogenase (*Mcad*); uncoupling protein 3 (*Ucp3*); and acyl-CoA thioesterase 1 (*Acot1*)), together with the TCA cycle (citrate synthase (*Cs*)) and ketone metabolism (3-hydroxy-3-methylglutaryl-CoA synthase 2 (*Hmgcs2*)), were also significantly increased (Figure 3B–G). However, only the change in MCAD expression was confirmed at the protein level (Figure 3C). There was no change in proteins involved in ATP synthesis or mitochondrial biogenesis/fission-fusion (Figure 4).

In hearts of *LAV-BPIFB4*-treated *db/db* mice, several enzymes were up-regulated at the mRNA level as compared with the PBS vehicle-treated diabetic group. These included the mitochondrial enzymes, *Pdk4*, *Hmgcs2*, *Cpt1b*, *Mcad*, *Ucp3*, *Acot1*, *Cs* and the B subunit of ATP-synthase (*Atpb*) (Figure 3). Furthermore, peroxisome proliferator-activated receptor γ co-activator 1 alpha (*Ppargc1α*), a transcriptional co-activator of mitochondrial biogenesis, together with the fusion protein, mitofusin 2 (*Mfn2*), and the regulator of mitochondrial transcription and mitochondrial DNA replication/repair, mitochondrial transcription factor A (*Tfam*), were also increased (Figure 4). Importantly, several these genes, namely HMGCS2 (Figure 3B), UCP3 (Figure 3E) and CS (Figure 3G), were also increased by *LAV-BPIFB4* at the protein level.

Together, these data indicate that hearts of *db/db* mice have limited alterations in metabolic enzyme expression compared with non-diabetic mice but that *LAV-BPIFB4* therapy is associated with subtle increases in mitochondrial-related enzymes. A schematic representation of these changes is provided in Figure 5.

## 4. Discussion

Further characterisation and understanding of available animal models of diabetic cardiomyopathy will be a significant factor in helping to facilitate the development of novel therapies [11,16]. In this study, we performed a multi-omics analysis on hearts of type-2 diabetic *db/db* mice to further characterise changes associated with the decline in cardiac function, and investigated the impact of transferring a gene variant already linked to prolonged health status in human populations [25].

In 13-week old type-2 diabetic *db/db* mice, altered cardiac lipid metabolism was the most prominent metabolic alteration observed from the metabolic phenotyping analysis. This included an elevation in metabolites within the acyl-carnitine, ceramides and sphingolipid, diacylglycerol, fatty acid, and triacylglycerides lipid classes. The intracellular accumulation of metabolites within some of these lipid classes has been associated with a disruption in mitochondrial function and suggested to exert toxic cellular effects, including interference of insulin-sensitive signalling pathways [26,27,28,29]. We also found several changes in membrane lipid metabolites within the glycerophospholipid and cardiolipin lipid classes, suggesting alterations in membrane composition, including that of the mitochondrial membrane, which can also have profound effects on cardiac function [30,31,32,33]. Cardiolipins, for example, have been shown to be vital for efficient functioning of the mitochondrial electron transport chain, and disruption in cardiolipin content leads to cardiac inefficiency and reduced contractility [32]. Changes in wide-ranging lipid species within the hearts of type-2 diabetic *db/db* mice, including some of those found within the present study, has also been reported previously [14]. Moreover, these changes were associated with significant alterations in sarcomere re-arrangement and loss of mitochondrial cristae and matrix volume [14]. It is also worth noting that similar changes in lipid metabolism has been reported in animal models of type-1 diabetes [34,35], suggesting some commonality between the different forms of diabetes.

Interestingly, more subtle changes were observed in the heart metabolome of diabetic *db/db* mice treated with *LAV-BPIFB4*, and three major outcomes can be appreciated: (1) the changes induced by treatment involved mitochondrial and membrane-related lipids, such as cardiolipins, acyl-carnitines and glycerophospholipids; (2) a significant percentage of the change (51 out of 63 metabolites = 81%) associated with *LAV-BPIFB4* therapy were unique to the treatment and independent of diabetic status; and (3) of the altered metabolites that were common to both the *LAV-BPIFB4* treated and untreated diabetic groups (12 out of 63 metabolites = 19%), only three became closer to the non-diabetic phenotype following treatment. Such changes may be linked to improved mitochondrial efficiency that could be a contributing factor in the observed SDF-1/CXCR4-dependent beneficial effects of *LAV-BPIFB4* on cardiac contractility in diabetic *db/db* mice [9]. The SDF-1/CXCR4 signalling axis has been suggested to positively modulate mitochondrial metabolism and function in isolated cells [36,37,38], and therefore may represent a central mechanistic pathway for *LAV-BPIFB4*-mediated cardioprotective effects. However, whether the SDF-1/CXCR4 signalling axis is involved in the metabolic effects observed in the present study requires further investigation.

It has been reported that alterations in metabolism is linked with changes in the expression profile of metabolic enzymes [39,40,41]. To obtain a wider profile of transcriptional changes occurring within the heart of type-2 diabetic *db/db* mice, we performed RNA-seq analysis which revealed an impact of diabetes on the heart transcriptome, including changes in metabolism- and immune-related genes. Several of these metabolic transcriptional changes were also confirmed by our targeted RT-qPCR analysis, with similar metabolic gene changes also being reported in a study conducted in type-1 diabetic Ins2^+/−^ Akita mice [42]. However, unlike experiments using type-1 diabetic mice, we did not observe any change due to type-2 diabetes at the protein level, suggesting a limited role for altered metabolic enzyme expression at this particular timepoint in the development of diabetic cardiomyopathy in *db/db* mice.

Similar to the metabolic phenotyping analysis, *LAV-BPIFB4* therapy also had an impact on the heart transcriptome, with the Principal Component Analysis of RNA-seq data emphasising that the expression profile of the *LAV-BPIFB4*-treated diabetic heart deviates further from that of the non-diabetic heart than that of the vehicle-treated diabetic group. Indeed, many of the changes observed between *LAV-BPIFB4*-treated and vehicle-treated diabetic mice involved a further up-regulation in genes related to metabolic processes, but unlike the untreated diabetic hearts, western blotting studies confirmed several mitochondrial-related genes also being up-regulated at the protein level. These included HMGCS2, CPT1b, UPC3 and CS, which could have positive effects on improving mitochondrial fatty acid handling and energy production in the heart of *LAV-BPIFB4*-treated mice. The upregulation of protein expression could result in improved function or compensation of impaired protein function.

In conclusion, this study contributes to the wider characterisation of the molecular and metabolic changes occurring in the type-2 diabetic *db/db* model of diabetic cardiomyopathy. A limitation of this comparison consists of not having used ND controls given the viral vector. Nonetheless, we have shown that the cardiac phenotype of ND mice is not altered by the AAV-LAV, thus suggesting that the transcriptomic data reflect a real difference between ND and diabetic mice. In addition, this is the first study to demonstrate that the transfer of a gene variant associated with longevity can confer a molecular and metabolic adaptation that could be a factor in supporting cardiac health. This was primarily associated with a relatively modest effect on mitochondrial-related pathways, suggesting that boosting mitochondrial function may be partially responsible for the cardiac protection afforded by forced expression of the longevity gene [9]. These findings call for an in-depth investigation of the mitochondrial metabolic network to determine its contribution in the maintenance of cardiac health in long-living individuals and the possible therapeutic exploitation of *LAV-BPIFB4* gene transfer for the treatment of diabetic cardiomyopathy.

## 5. Patents

AAP and CV have filed a patent on the potential therapeutic use of LAV-BPIFB4.

## Figures and Tables

**Figure 1 cells-09-01283-f001:**
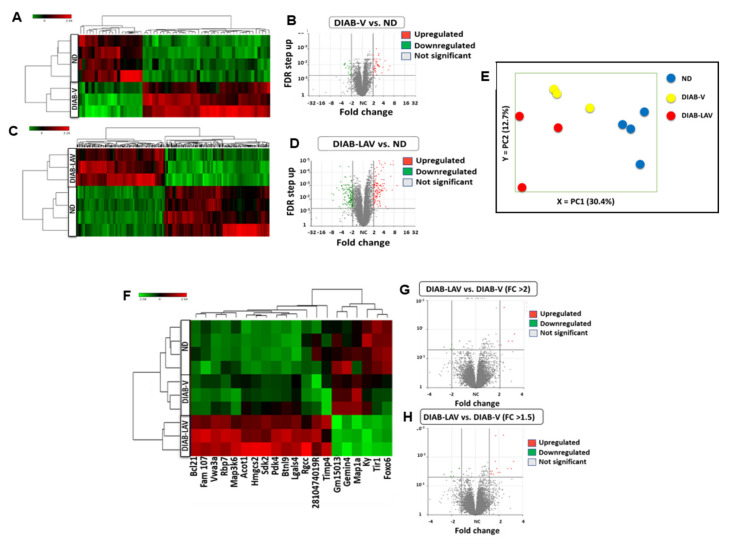
RNA sequencing. (**A**) Cluster analysis and (**B**) volcano plot in the contrast between type-2 diabetic *db/db* mice given PBS vehicle (DIAB-V) and untreated ND controls. (**C**) Cluster analysis and (**D**) volcano plot in the contrast between *LAV-BPIFB4*-treated diabetic *db/db* mice (DIAB-LAV) and ND controls. (**E**) Principal component analysis emphasising the variation in gene expression between the different groups. (**F**) Cluster analysis and (**G**,**H**) volcano plots in the contrast between type-2 diabetic *db/db* mice given *LAV-BPIFB4* or PBS vehicle at >2.0 FC or >1.5 FC level. *n* = 3 to 4 mice per group.

**Figure 2 cells-09-01283-f002:**
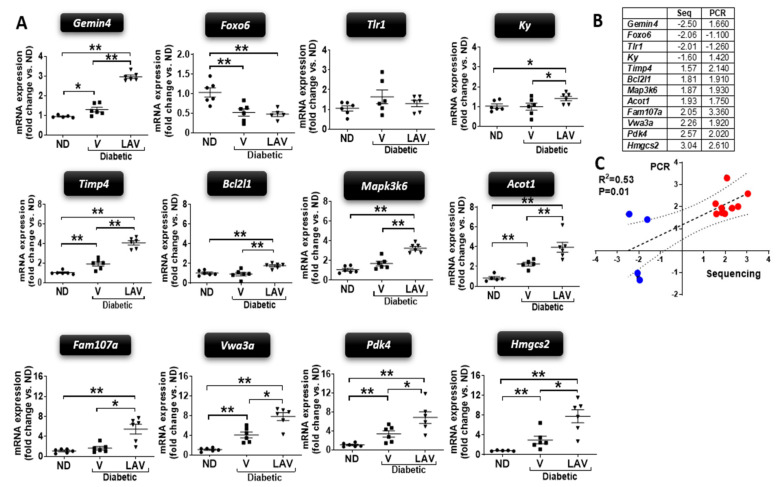
Validation of sequencing data by RT-qPCR (Study 2). (**A**) Graphs showing the mRNA levels of 12 genes found differentially modulated by the *LAV-BPIFB4* treatment in type-2 diabetic *db/db* mouse hearts. RT-qPCR was performed on the samples from sequencing, plus additional animals, for an *n* = 6 animals per group (non-diabetic, ND; PBS vehicle-treated diabetic, V; and *LAV-BPIFB4*-treated diabetic, LAV). Data (fold changes vs. ND) are expressed as individual values and mean ± SEM. * *p* < 0.05 and ** *p* < 0.01. (**B**) Comparison of fold changes from the two methodologies. (**C**) Linear regression indicating the correlation of data from the two methodologies.

**Figure 3 cells-09-01283-f003:**
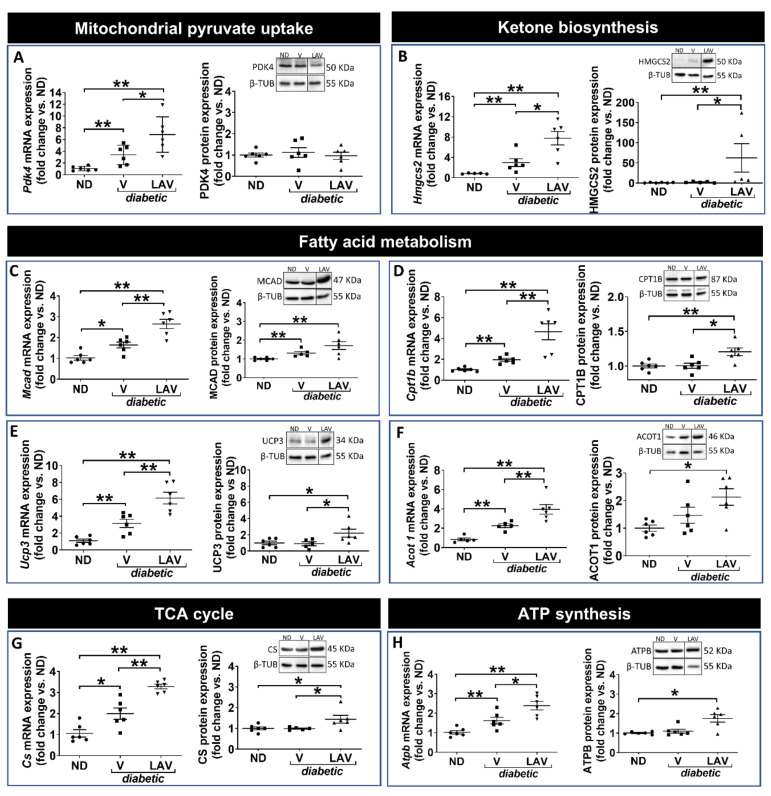
Focused transcriptomic analysis of mitochondrial metabolic targets. Graphs show the mRNA and protein expression levels of components involved in the regulation of mitochondrial pyruvate uptake (PDK4) (**A**); ketone biosynthesis (HMGCS2) (**B**); fatty acid metabolism (MCAD, CPT1B, UCP3, & ACOT1) (**C**–**F**); the TCA cycle (CS) (**G**); and ATP synthesis (ATPB). Data (fold changes vs. ND) are expressed as individual values and means ± SEM of *n* = 6 per group. * *p* < 0.05 and ** *p* < 0.01. Representative western blots show n=1 sample per group, presented in the same order as in the graphs. Line within blots indicates non-adjacent lanes on the same membrane (see online Appendix A). For clarity, mRNA data for *Pdk4*, *Hmgcs2* and *Acot1* is the same as that presented in Figure 2.

**Figure 4 cells-09-01283-f004:**
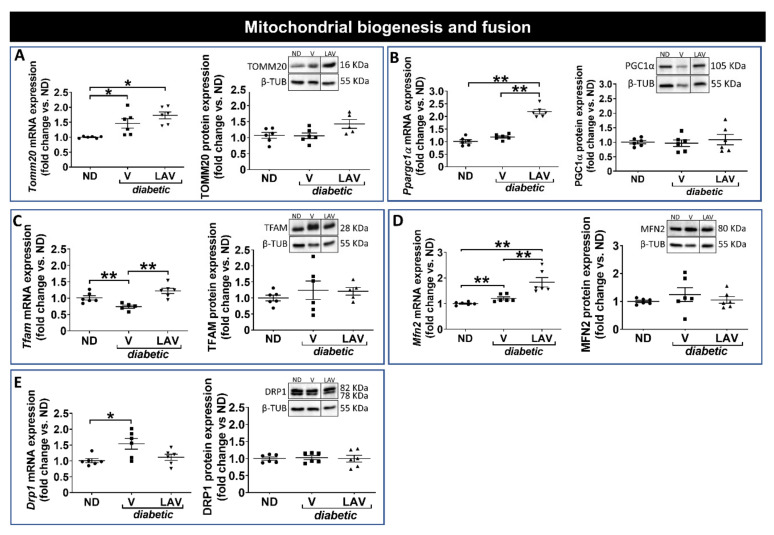
Focused transcriptomic analysis of mitochondrial biogenesis and dynamics. Graphs show the mRNA and protein expression levels of components involved in mitochondrial protein import (TOMM20) (**A**); biogenesis (PCG1α and TFAM) (**B**,**C**); and fussion/fission (MFN2 and DRP1) (**D**,**E**). Data (fold changes vs. ND) are expressed as individual values and means ± SEM of *n* = 6 per group. * *p* < 0.05 and ** *p* < 0.01. Representative western blots show n = 1 sample per group, presented in the same order as in the graphs. Line within blots indicates non-adjacent lanes on the same membrane (see online Appendix A).

**Figure 5 cells-09-01283-f005:**
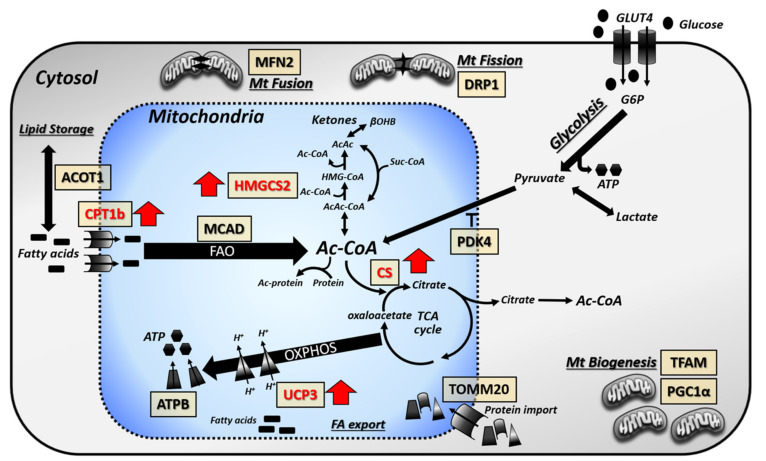
Schematic representation of metabolic protein changes induced by *LAV-BPIFB4*. *LAV-BPIFB4* treatment is associated with a subtle up-regulation in proteins involved in mitochondrial metabolism. Red arrows indicate the enzymes induced by *LAV-BPIFB4* treatment.

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
