# Peer review of "Multi-Omics Analysis of Diabetic Heart Disease in the db/db Model Reveals Potential Targets for Treatment by a Longevity-Associated Gene"

_cells, 2020, doi:10.3390/cells9051283_

Round 1
Reviewer 1 Report
In this study, Faulkner et al employ proteomic and transcriptomic techniques to investigate mechanisms underlying diabetic cardiomyopathy in the db/db mouse model. The authors had previously demonstrated LAV-BPIFB4 to be protective and through AAV transduction, test its influence on the above. While somewhat descriptive, some findings are surprising and the study does shed important light on mechanisms of diabetic cardiomyopathy. I have listed several concerns below that would need to be addressed prior to publication:
- While immunohistochemistry confirms expression in cardiac tissue, how specific was AAV9 gene transduction to the heart?
- It appears RT-qPCr was performed on left ventricles but this is not indicated for RNA-seq/protein extraction for western and mass spectrometry – were these whole heart?
- Vehicle is mentioned to be either AAV-GFP or 100 ul PBS, this needs clarification in results section and legends as to when each was used.
- In Figure 3 each lane of western blots should be labeled for clarification. Why are they spliced? Are images from the same membrane? Tubulin seems quite variable in H especially. Please endure all bands from westerns presented are from the same membrane and same exposure for appropriate interpretation.
- Please expand discussion on how the function of LAV-BPIFB4 relates to observed metabolic changes.
- Typo in Figure 3B where HMGCS2 graph is labeled ‘rotein’
Reviewer 2 Report
The authors investigated the influence of the adenoviral transfer of the longevity-associated variant (LAV) of the human BPIFB4 gene on the cardiac metabolism and transcriptome in diabetic db/db mice in comparison to untreated non-diabetic wt/db mice. Ultra-high-performance liquid chromatography-mass spectrometry and next-generation RNAseq were used. The results were verified by PCR and Western blots. Combining analyses of metabolism and gene expression is an interesting approach to understand processes in diseased hearts. However, I have a major concern.
The authors used untreated non-diabetic wt/db mice (ND) as controls. The adequate control would have been non-diabetic wt/db mice, which were treated with an adenoviral control-vector, as done for diabetic mice. Adenovirus infection has definitely an effect on the heart tissue. When comparing untreated ND with vector-treated diabetic mice, which changes are related to the vector and which to diabetes? That is especially important because viral infection and activated immune response by the virus takes significant influence on the mitochondrial function and the energy metabolism of the heart. Thus, only the comparison between vector-treated and LAV-BPIFB4-treated diabetic mice is reliable. I recommend to remove the ND group from the manuscript or use vector-treated ND mice as control. You should also find a better way to present the essential mRNA expression data in the manuscript. It is difficult to work through all the tables. It is especially confusing when expression is presented as “upregulated” in the text but you find a fold change <-1 (compare table S5 and page 5, line 245). Please, adapt this to each other.
Please correct in table S3 line 5, “higher in GPF” to “higher in LAV”.
You should introduce BPIFB4 and LAV-BPIFB4 in more detail in the introduction.
Please insert “vehicle-treated” diabetic db/db mice….., page 5, line 245.
You showed the cardiac expression of LAV-BPIFB4 by immunostaining. Did the expression spread over the whole heart or was it focally located?
Which statistical conditions did you use in the GOslim analysis? GOslim results should be shown for table 7 or 8.
The expression of genes/proteins that you demonstrate in figure 3 and 4 are important for the mitochondrial function but not all of them are found in your mRNA expression data (e.g. Mcad, CS, Cpt1…). Why did you choose especially these proteins. Why did you not use more of the components whose mRNA expression was modified to support your transcriptome data?
Page 11, line 401: The upregulation of protein expression is not always a sign of improved function. In contrast, it also occurs as a compensating effect of impaired protein function.
You uploaded your supplemented data as “supplements” and as “unpublished data”. I assume that you wanted to attach your manuscript (9) that is in press (?).
Round 2
Reviewer 2 Report
Still, I would like to get more information about BPIFB4 in the introduction. This helps the readers who are not familiar with BPIFB4 to get into the function of this protein. For instance, what kind of protein is it, where is it expressed, where is it localized, what is its function. The reader must collect these information from other sources. A better understanding of BPIFB4 would arouse more interest in your study.
Author Response
We have further detailed the characteristics of the studied gene in Introduction (line 59 forward)
A genome-wide association study on LLIs of three different geographic areas identified a Longevity Associated Variant (LAV) of the bactericidal/permeability‐increasing fold‐containing family B member 4 (BPIFB4), a four‐missense single‐nucleotide polymorphism haplotype allele. BPIFB4 belongs to a family of proteins with lipid binding pockets involved in many functions, from anti-microbial (BPI) to cholesterol handling (CETP). Specifically, LAV confers novel functions to BPIFB4 protein, which becomes able to activate eNOS via a SDF1/CXCR4/calcium mediated mechanism. Additionally, we have described a unique therapeutic effect of the LAV isoform as compared to the wild type (WT). Indeed, delivery of the LAV, but not the WT form of BPIFB4 gene effectively contrasts the development of atherosclerosis and benefits the recovery from ischemia potentiating endothelial function [7,8]. These effects are blunted by CXCR4 inhibitors and are associated with a monocyte skewing toward an M2 phenotype.